# Finite-Key Analysis for Quantum Key Distribution with Discrete-Phase Randomization

**DOI:** 10.3390/e25020258

**Published:** 2023-01-31

**Authors:** Rui-Qiang Wang, Zhen-Qiang Yin, Xiao-Hang Jin, Rong Wang, Shuang Wang, Wei Chen, Guang-Can Guo, Zheng-Fu Han

**Affiliations:** 1CAS Key Laboratory of Quantum Information, University of Science and Technology of China, Hefei 230026, China; 2CAS Center for Excellence in Quantum Information and Quantum Physics, University of Science and Technology of China, Hefei 230026, China; 3Hefei National Laboratory, University of Science and Technology of China, Hefei 230088, China; 4State Key Laboratory of Cryptology, Beijing 100878, China; 5Department of Physics, University of Hong Kong, Pokfulam, Hong Kong

**Keywords:** quantum key distribution, finite-key analysis, discrete-phase randomization

## Abstract

Quantum key distribution (QKD) allows two remote parties to share information-theoretic secret keys. Many QKD protocols assume the phase of encoding state can be continuous randomized from 0 to 2π, which, however, may be questionable in the experiment. This is particularly the case in the recently proposed twin-field (TF) QKD, which has received a lot of attention since it can increase the key rate significantly and even beat some theoretical rate-loss limits. As an intuitive solution, one may introduce discrete-phase randomization instead of continuous randomization. However, a security proof for a QKD protocol with discrete-phase randomization in the finite-key region is still missing. Here, we develop a technique based on conjugate measurement and quantum state distinguishment to analyze the security in this case. Our results show that TF-QKD with a reasonable number of discrete random phases, e.g., 8 phases from {0,π/4,π/2,…,7π/4}, can achieve satisfactory performance. On the other hand, we find the finite-size effects become more notable than before, which implies that more pulses should be emit in this case. More importantly, as a the first proof for TF-QKD with discrete-phase randomization in the finite-key region, our method is also applicable in other QKD protocols.

## 1. Introduction

Quantum key distribution (QKD) [1,2],one of the most successful and mature applications in quantum information science, allows for two legitimate parties (Alice and Bob) to share information-theoretic secret keys. In theory, its security has been proved [3,4,5], while experiments towards a higher key rate [6] and longer achievable distance [7,8,9,10] have been demonstrated. Still, some large scale QKD networks are emerging [11,12,13,14]. However, owing to the inherent photon-loss in the channel, it meets a vital bottleneck that limits the communication distance and key generation rate. Specifically, some fundamental rate-loss limits [15,16] impose a restriction on any point-to-point QKD without repeaters. More precisely, the key rate *R* is bounded by the channel transmission probability η with the linear PLOB bound $R=-\log_{2}\left(1-\eta \right)$ [16]. Delightfully, M.Lucamarini et al. made a breakthrough by proposing twin-field (TF) QKD in 2018. The essential idea of TF-QKD is in code mode extracting the key bit from a single-photon click event of the measurement station located in the middle of channel, which happens with a probability proportional to $\sqrt{\eta }$; thus, surpassing the linear PLOB bound becomes possible, and a so-called phase-error rate may be estimated in decoy mode [17,18,19] to monitor security. Driven by this, several TF-type QKD protocols [20,21,22,23,24,25,26] were proposed later to complete security proofs and improve performance. Based on these protocols, experimentalists also made great efforts to realize TF-QKD [27,28,29,30,31,32,33,34,35].

Since TF-QKD inherits measurement-device-independent (MDI)-QKD’s [36] merit that is immune to all side-channel attacks to measurement devices and all measurement-device imperfections [37,38], one does not need to take the detection loopholes into account within the TF-type QKD system. In spite of this, the security issues of the state preparation in TF-QKD must be carefully considered. In practice, the laser source of TF-QKD is usually a continuous source emitting coherent states with a fixed phase. Meanwhile, continuous phase-randomization from 0 to π is required in the TF-QKD. More specifically, this continuous phase-randomization is assumed in both the code and test modes in Refs. [21,23,26], or at least in the test mode in Refs. [22,24,39]. To fulfill this requirement, Alice and Bob must randomize the global phase continuously and uniformly. Unluckily, two ways to achieve phase randomization introduce different problems in the experiment. Passive randomization will lead to phase correlations between adjacent pulses [40,41], while active randomization can only randomize the phase over discrete set of values.

To bridge this gap between theory and experiment, two works that analyzed the security of fully discrete-phase randomization TF-QKD protocol have been proposed [42,43,44] in these days. However, a security proof in the finite-key region is still missing. Hence, one natural question is that whether TF-QKD with fully discrete-phase randomization can work well non-asymptotically. This work affirms that it can.

In this paper, we analyze the security of TF-QKD protocol with fully discrete randomization in a finite-key region. Interestingly, our analysis leads to comparable performance with the continuous one. Since taking the discrete phase into account, our results make the TF-QKD more practical and can be applied to the future TF-QKD experiment. More importantly, some techniques proposed here, e.g., Lemma A1 (introduced later), can be utilized to analyze the security of other QKD protocols with discrete-phase randomization.

This work is organized as follows. In Section 2, we give a description of the TF-QKD protocol with fully discrete-phase randomization, and the sketch of the security proof is given in Section 3. Note that the proof is detailed in Appendix A. In Section 4, by the numerical simulation, we show this protocol can still beat the linear PLOB bound [16] and has satisfactory performance. Finally, a conclusion is given in Section 5.

## 2. Protocol Description

Indeed, the protocol analyzed here has been depicted in Ref. [43]. For ease of understanding, we illustrate the protocol as follows.

Step 1: Alice (Bob) chooses a label from {“μ”,“0”,“ν”} with probabilities $P_{\mu },P_{O},P_{\nu }$, according to the label she (he) chooses, she (he) takes one of the following actions:

“μ”: she (he) randomly picks an integer $l_{A_{c}}$ ($l_{B_{c}}$) from {0,1,⋯,M−1} with equal probability $\frac{1}{M}$ where M is an even integer. This means that the phase 2π is divided into M parts. Then, she (he) randomly chooses a key bit $k_{a}$$\left(k_{b}\right)$ where $k_{a}\left(k_{b}\right)\in \left\{0,1\right\}$. Finally, she (he) sends a pulse with a coherent state $|e^{i(\frac{l_{Ac}}{M}2\pi +\pi k_{a})}\sqrt{\mu }\rangle$($|e^{i(\frac{l_{B_{c}}}{M}2\pi +\pi k_{b})}\sqrt{\mu }\rangle$ ).

“0”: she (he) sends the vacumm state.

“ν”: she (he) randomly picks an integer $l_{A_{c}}$ and $l_{B_{c}}$ from {0,1,⋯,M−1} with equal probability $\frac{1}{M}$ where M is an even integer. This means that the phase 2π is divided into M parts. Then, she (he) sends a pulse with a coherent state $|e^{i\frac{l_{Ac}}{M}2\pi }\sqrt{\mu }\rangle$($|e^{i\frac{l_{Bc}}{M}2\pi }\sqrt{\mu }\rangle$ ).

The first case is called code mode, while the other cases are decoy mode.

Step 2: Alice and Bob repeat Step 1 in total of $N_{tot}$ times.

Step 3: After receiving $N_{tot}$ pairs of pulses from Alice and Bob, interfering each pair at a beamsplitter and measuring the two outputs with his single photon detectors (SPDs), an honest Eve announces whether or not each measurement is successful. Here, ’successful’ means only one SPD (left SPD or right SPD) clicks in the corresponding measurement, and if so, Eve reports the specific SPD clicked.

Step 4: For those rounds Eve announcing successful click, Alice and Bob announce the intensities they chose as well as the values of $l_{A_{c}}$ and $l_{B_{c}}$. Then, Alice and Bob only retain those successful rounds in which the intensities of the coherent state they sent are same while the in-phase ($l_{A_{c}}=l_{B_{c}}$) or anti-phase ($|l_{A_{c}}-l_{B_{c}}|=M/2$) condition is also met. Let $n_{2\beta }^{+}$($n_{2\beta }^{-}$) be the number of the retained rounds when both Alice and Bob chose the same intensity β of the coherent state and the in-phase (anti-phase) is also met. Note that we assume $l_{A_{c}}=l_{B_{c}}=0$ always holds in the case of β=0. Alice and Bob generate their sifted keys from $n_{2\mu }=n_{2\mu }^{+}+n_{2\mu }^{-}$ retained rounds in code mode, thus the length of sifted key bits $n_{bit}=n_{2\mu }$. Note that if it is an in-phase (anti-phase) round with right (left) SPD clicking, Bob may flip his corresponding sifted key bit.

Step 5: With all of the quantities $n_{2\beta }=n_{2\beta }^{+}+n_{2\beta }^{-}$, Alice and Bob use linear programming to obtain an upper bound on the number of phase errors(defined later) $n_{ph}^{U}$ with a failure probability no more than ε; then, they can calculate the upper bound $e_{ph}^{U}=n_{ph}^{U}/n_{bit}$.

Step 6: Step 6 consists of error correction and privacy amplification.

Step 6a: Alice sends $H_{EC}$ bits of syndrome information of her sifted key bits to Bob through an authenticated public channel. Then, Bob uses it to correct errors in his sifted keys. Alice and Bob calculate a hash of their error-corrected keys with a random universal hash function and check whether they are equal. If equal, they continue to the next step; otherwise, they abort the protocol.

Step 6b: Alice and Bob apply the privacy amplification to obtain their final secret keys. If the length of their secret key satisfies $l=n_{bit}\left(1-h\left(e_{ph}^{U}\right)\right)-H_{EC}-\log_{2}\frac{2}{\epsilon_{cor}}-\log_{2}\frac{1}{4\epsilon_{PA}^{2}}$ where h(·) denotes the binary Shannon entropy, this protocol must be $\epsilon_{cor}$-correct and $\epsilon_{sec}$-secret with $\epsilon_{sec}=\sqrt{\varepsilon }+\epsilon_{PA}$. Here, $\epsilon_{cor}$($\epsilon_{sec}$) represents the protocol is correct (secret) with a failure probability no more than $\epsilon_{cor}\left(\epsilon_{sec}\right)$. Hence, the total security parameter is $\epsilon_{tol}$-secure where $\epsilon_{tol}=\epsilon_{cor}+\epsilon_{sec}$. It is elaborated thoroughly in the widely-used universally composable security framework [45,46].

## 3. Security Proof

In this section, we present the security proof of this protocol. The main task of the security proof is to bound the information Eve holds. To accomplish this task, one can calculate a so-called phase-error rate. Firstly, we construct an equivalent virtual protocol, in which Alice and Bob prepare some entangled states between local states and traveling states, but traveling states must have the same density matrices as actual protocol in the channel. The sifted key bits can be seen as the outputs of measurement with *Z*-basis on local states made by Alice and Bob; then, the so-called phase-error rate is defined as the error rate for the outputs of measurement with the *X*-basis made by them. According to the complementarity argument [47], the phase-error rate can be used to bound Eve’s information on the sifted keys. In the following, we give the virtual protocol and show how to bound the phase-error rate.

### 3.1. Equivalent Virtual Protocol

In our virtual protocol, Alice generates secret keys from the code mode in which she prepares the state
(1)$$\begin{array}{c} {|\psi \rangle }_{\mu ,A_{c}Aa}=\sum_{l=0}^{M-1}\frac{1}{\sqrt{M}}{|l\rangle }_{A_{c}}(\frac{1}{\sqrt{2}}{(|0\rangle }_{A}|e^{i\frac{2\pi }{M}l}\sqrt{\mu }\rangle_{a}+{|1\rangle }_{A}|-e^{i\frac{2\pi }{M}l}\sqrt{\mu }\rangle_{a})), \end{array}$$
where $A_{c}$ and A are the local quantum systems in Alice’s side, and *a* is the traveling quantum state Alice sent to Eve. Similarly, Bob prepares ${|\psi \rangle }_{\mu ,B_{c}Bb}$ defined analogously to ${|\psi \rangle }_{\mu ,A_{c}Aa}$. Obviously, Alice (Bob) measures A(B) with *Z*-basis to obtain sifted key, i.e., ${|0\rangle }_{A}$ for bit 0 and ${|1\rangle }_{A}$ for bit 1. In order to obtain the phase-error rate, they measure A,B in *X*-basis {|+〉,|−〉} after Eve’s attack. As for the test mode, we assume Alice prepares the following states
(2)$$\begin{array}{cc} {|\psi \rangle }_{0,A_{c}Aa} & ={|0\rangle }_{A_{c}}{|0\rangle }_{A}{|0\rangle }_{a},\\ {|\psi \rangle }_{\nu ,A_{c}Aa} & =\sum_{l=0}^{M-1}\frac{1}{\sqrt{M}}{|l\rangle }_{A_{c}}{|0\rangle }_{A}|e^{i\frac{2\pi }{M}l}\sqrt{\nu }\rangle_{a}. \end{array}$$
here, the local states of $A_{c}$ are encoded in photon-number states, and Alice can measure $A_{c}$’s photon-number to learn the phase of sent states.

Finally, we can describe the process of state preparation above with a single state, namely,
(3)$$\begin{array}{c} {|\psi \rangle }_{A_{s}A_{c}Aa}=\sqrt{p_{\mu }}{|0\rangle }_{A_{s}}{|\psi \rangle }_{\mu ,A_{c}Aa}+\sqrt{p_{O}}{|1\rangle }_{A_{s}}{|\psi \rangle }_{0,A_{c}Aa}+\sqrt{p_{\nu }}{|2\rangle }_{A_{s}}{|\psi \rangle }_{\nu ,A_{c}Aa} \end{array}$$
where Alice’s additional local ancilla $A_{s}$ is in the photon number states. Similarly, Bob can prepare ${|\psi \rangle }_{B_{s}B_{c}Bb}$ defined analogously to ${|\psi \rangle }_{A_{s}A_{c}Aa}$. Though Alice (Bob) may measure $A_{s}$($B_{s}$),$A_{c}\left(B_{c}\right)$, and *A*(*B*) after or before Eve announcing her measurement results, Alice (Bob) must announce the measurement results after Eve’s announcement then post-select the successful rounds. The following is a detailed illustration of our equivalent virtual protocol.

Step 1:

Alice and Bob prepare a gigantic quantum state $|\Phi \rangle ={|\phi \rangle }^{⊗N_{tot}}={(|\psi \rangle }_{A_{s}A_{c}Aa}⊗{|\psi \rangle }_{B_{s}B_{c}Bb}{)}^{⊗N_{tot}}$ and send all subsystems *a* and *b* to Eve through an insecure quantum channel.

Step 2:

After performing an arbitrary quantum operation on all subsystems *a* and *b* from Alice and Bob, Eve announces whether it has a successful click (only one of her SPDs clicks) or not for each round. For a successful round, Eve continues to announce whether the left SPD clicks or the right SPD clicks. We use M($\bar{M}$) to denote the set of successful (unsuccessful) rounds.

Step 3:

For those rounds in which Eve announces success, Alice and Bob jointly measure the subsystem $A_{c}$($B_{c}$) and $A_{s}\left(B_{s}\right)$ in the photon-number basis to learn whether the intensities of the coherent state they send are same or not and whether it is in-phase or anti-phase. Then, they only retain those rounds where in-phase or anti-phase is met, and they choose the same intensities. Let $M_{s}$ denote the set of those retained rounds, while $M_{f}$ denotes those rounds that are in M but not in $M_{s}$.

Step 4:

For these rounds in $M_{s}$, Alice (Bob) measures the subsystem $A_{c}A_{s}$($B_{c}B_{s}$) in Fock basis to learn the phase and intensity of the coherent states she (he) sent. If the result of $A_{s}$($B_{s}$) is in state ${|0\rangle }_{A_{s}}$(${|0\rangle }_{B_{s}}$), she (he) measures subsystems *A*(*B*) in the Z basis to decide her (his) sifted key, respectively; otherwise, she (he) measures subsystem A(B) in the Z basis but does not incorporate these measurement outcomes in her (his) sifted key.

Step 5 to Step 6:

Let $n_{2\beta }$ be the number of rounds in $M_{s}$ satisfying that both Alice and Bob chose the intensity β. With parameters $n_{2\beta }$, perform the same operations as Step 5 to Step 6, respectively, in the actual protocol given in Section 2.

### 3.2. Estimation of Phase-Error Rate

The essential of security proof is to estimate the upper-bound of the phase-error rate $e_{ph}$ of the sifted keys, i.e., how many same or different outcomes Alice and Bob have if they measure *A* and *B* with *X*-basis hypothetically in the rounds where sifted keys are generated. Specifically, in our protocol, we define the number of the same outcomes they have as $n_{ph}$, i.e., the number of phase-error events. Provided that $e_{ph}=n_{ph}/n_{2\mu }$ is bounded, one can generate the final secret key with an appropriate $\epsilon_{tol}$ value as given in Step6.b of the actual protocol.

A detailed proof for how to estimate $n_{ph}$ is present in Appendix A. Here, a sketch of this proof is given.

Though analyzing the equivalent protocol, it is proven that if Alice and Bob both chose intensity β, and in-phase or anti-phase is also met, they actually prepare a mixture $\tau_{2\beta }$, which consists of component $\tau_{j|2\beta },j=0,1,...,M-1$. Moreover, each phase-error event is a click by some particular components of that mixture $\tau_{2\beta }$, i.e., $\tau_{j|2\beta },j=0,2,...,M-2$. These results imply that
(4)$$\begin{array}{cc} n_{2\mu }= & \sum_{j=0}^{M}n_{j|2\mu },\\ n_{2\nu }= & \sum_{j=0}^{M}n_{j|2\nu },\\ n_{ph}= & \sum_{j=0,j\in N_{0}}^{M-2}n_{j|2\mu }, \end{array}$$
where $n_{j|2\beta }$ denotes the number of rounds in $M_{s}$, in which Alice and Bob both chose intensity β, but $\tau_{2\beta }$ is actually $\tau_{j|2\beta }$. Meanwhile, $N_{0}$ is the set of even numbers. Now, the hypothetical value $n_{ph}$ is related to some experimentally observed values. However, just with these equations it is difficult to bound $n_{ph}$ tightly since $n_{j|2\beta }$ cannot be known directly.

On the other hand, both $\tau_{j|2\mu }$ and $\tau_{j|2\nu }$ are very close to Fock-state |j〉〈j|. Accordingly, it is intuitive to consider if there are constraints on the gap between $n_{j|2\mu }$ and $n_{j|2\nu }$. Then, we developed Lemma A1 (see Appendix A for details) to bound the gap between the yields of two distinct quantum states in a non-asymptotic situation. Applying this lemma, we obtained a series of constraints on $n_{j|2\mu }$ and $n_{j|2\nu }$. Finally, combined with Equation (Equation 4), an analytical upper bound of $n_{ph}$ (given in the end of the Appendix A) was calculated to find the upper bound of phase-error rate $e_{ph}^{U}=n_{ph}^{U}/n_{2\mu }$.

## 4. Numerical Simulation

In this section, we simulate the final secret key rate with the parameters listed in Table 1.

It is reasonable to simulate the experimentally observed values $n_{2\mu }$, $n_{2\nu }$ and $n_{0}$ with their mean values. Let $Q_{corr|2\beta }$ be the probability of only one click from left (right) SPD when both Alice and Bob prepare coherent states with intensity β and a phase difference of 0 (π), and $Q_{err|2\beta }$ be the probability of only one click from left (right) SPD when both Alice and Bob prepare coherent states with intensity β and phase difference of π (0). Then, we have
(5)$$\begin{array}{c} Q_{corr|2\beta }=\left(1-\left(1-p_{d}\right)e^{-2\eta (1-e_{m})\beta }\right)e^{-2\eta e_{m}\beta }\left(1-p_{d}\right),\\ Q_{err|2\beta }=\left(1-\left(1-p_{d}\right)e^{-2\eta e_{m}\beta }\right)e^{-2\eta (1-e_{m})\beta }\left(1-p_{d}\right), \end{array}$$
where $\eta =10^{\frac{-0.2L}{20}}$ and *L* is the channel distance between Alice and Bob. Accordingly, in the simulation, we assume $n_{2\beta }=N_{tot}P_{\beta }^{2}2\left(Q_{corr|2\beta }+Q_{err|2\beta }\right)/M$ for β=μ,ν. Note that $n_{0}=N_{tot}P_{O}^{2}\left(Q_{corr|0}+Q_{err|0}\right)$, $n_{bit}=n_{2\mu }$ and $e_{bit}=Q_{err|2\mu }/\left(Q_{corr|2\mu }+Q_{err|2\mu }\right)$. With these values, setting M=8 and the failure probability of estimating phase error $\varepsilon =\left(6M+12\right)\varepsilon_{a}=4\times 10^{-20}$, one can obtain the upper-bound of phase-error rate $e_{ph}^{U}$ by the linear programming given by (Equation 46) in Appendix A. Moreover, the amount of $H_{EC}$ is $H_{EC}=N_{bit}fh\left(e_{bit}\right)$, $\epsilon_{cor}=1\times 10^{-10}$, and $\epsilon_{PA}=1.6566\times 10^{-10}$, which leads to a secret key of length $l=n_{bit}\left(1-h\left(e_{ph}^{U}\right)\right)-H_{EC}-\log_{2}\frac{2}{\epsilon_{cor}}-\log_{2}\frac{1}{4\epsilon_{PA}^{2}}$ with $\epsilon_{sec}=\epsilon_{PA}+\sqrt{\varepsilon }$ and the total security parameter $\epsilon_{tol}=\epsilon_{cor}+\epsilon_{sec}=4.6566\times 10^{-10}$.

Finally, we numerically optimize the intensities and corresponding probabilities to maximize *l* in the cases of the total number of pulses is $N_{tot}=1\times 10^{17},1\times 10^{14},1\times 10^{13},1\times 10^{12}$. Note that because this numerical problem is very time-comsuming, these intensities and probabilities are not optimized at each distance. Additionally, we use some typical parameters instead. The simulate results ($l/N_{tot}$ v.s. L) are illustrated below.

As Figure 1 shows, we obtain considerable secret key rates when the total number of pulses is $10^{12}$, $10^{13}$, $10^{14}$ or $10^{17}$. Through numerical simulations, it is confirmed that TF-QKD with discrete-phase randomization has satisfactory performance. On the other hand, it is verified that finite-size effects become more notable here compared with the original protocol with continuous phase randomization; it seems that one has to prepare $10^{17}$ pulses to surpass the PLOB linear bound. This is because the statistical fluctuations in Lemma A1 are proportional to the square root of the total number of emitting pulse $N_{tot}$, which leads to alarge phase-error rate $e_{ph}$ when $n_{bit}$ is not sufficiently large.

## 5. Conclusions

In real setups of TF-QKD, continuous randomization is usually realized by actively adding a random signal to a phase modulator. On the other hand, random numbers are generated discretely in most schemes. Therefore, TF-QKD with discrete-phase randomization is more practical. It is necessary to analyze the security of TF-QKD with discrete-phase randomization. Based on conjugate measurement, the security proof of a QKD protocol is to estimate the phase-error rate. Then in case of discrete-phase randomization, a critical step is knowing how to bound the gap between yields of two distinct but very close quantum states in a non-asymptotic situation. To achieve this goal, Lemma A1 is developed to find the upper bound of this gap. With the help of Lemma 1, linear programming is proposed to calculate the phase-error rate, and the key length is then straightforward. Through numerical simulations, it is confirmed that TF-QKD with discrete-phase randomization has satisfactory performance. On the other hand, we also find that more pulses should be prepared to alleviate the finite-size effects than previous protocol.

Moreover, it is worth noting that Lemma A1 is quite useful in a variety of scenarios, not just in the security proof of TF-QKD. For instance, if one considers the BB84 with discrete-phase randomization [48], the Lemma A1 can be utilized to bound the yield of single photon state, so then it is not difficult to give a relevant security proof. To summarize, we give the first security proof for TF-QKD with finite discrete-phase randomization in non-asymptotic scenarios. Although the proof is tailored for TF-QKD, the framework of this proof, i.e. Lemma A1, can be adapted in other protocols.

## Figures and Tables

**Figure 1 entropy-25-00258-f001:**
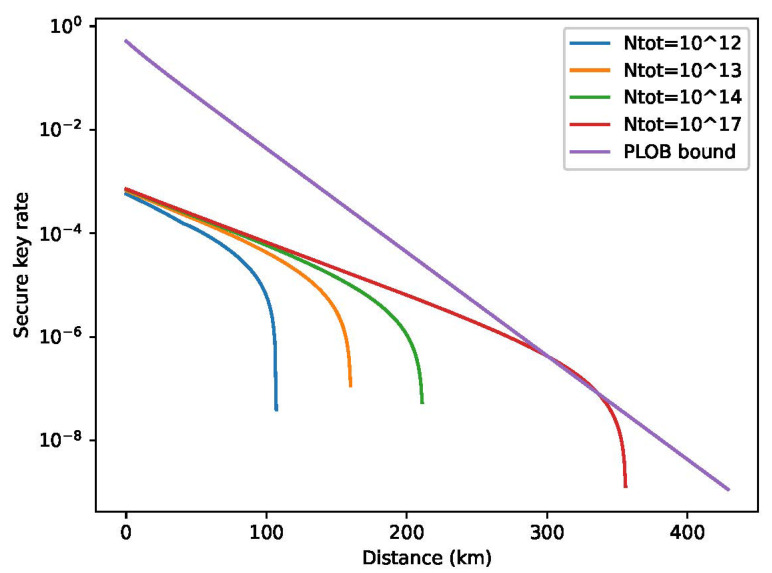
Secret key rate ($l/N_{tot}$) of fully discrete TF-QKD [43]. In this figure, the key rate corresponding to the total number of pulses $N_{tot}$ is $1\times 10^{12},1\times 10^{13},1\times 10^{14},1\times 10^{17}$, plotted above. Note that we set M=8 in the simulation.

**Table 1 entropy-25-00258-t001:** List of parameters uesd in the numerical simulations. Here, $e_{m}$ is loss-independent misalignment error rate due to optical imperfect interference, $p_{d}$ is dark counting probability for each SPD, ξ is fiber loss constant, $\eta_{d}$ denotes detection efficiency of each SPD, *f* is error-correction inefficiency, and $\epsilon_{tol}$ denotes the total security coefficient.

$e_{m}$	$p_{d}$	ξ (dB/km)	$\eta_{d}$	*f*	$\epsilon_{\mathrm{tol}}$
0.03	$1\times 10^{-8}$	0.2	0.3	1.1	$4.6566\times 10^{-10}$

## Data Availability

Not applicable.

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
