# Peer review of "Finite-Key Analysis for Quantum Key Distribution with Discrete-Phase Randomization"

_entropy, 2023, doi:10.3390/e25020258_

Round 1

Reviewer 1 Report

In some important QKD protocols, e.g. decoy-states BB84 and TF-QKD, the global phase of each quantum state is assumed to be randomized continuously. This assumption plays an essential role for the security proofs. Nevertheless, it’s questionable in some implementations. As discussed in this work, the TF-QKD may have to perform randomize this phase with a finite set of phases, i.e. discrete phase randomization. This work focuses on this case and finite-key effect is also considered. It seems to be the first security proof taking both discreet phase randomization and finite-key into consideration, thus I can recommend its publication in principle. However, I think some issues must be clarified before publication.

1.      The lemma1 is the core of the security proof. Although the proof seems correct, some points are misleading or not easy to understand. In the optimal attack, the probability of guessing correctly is a constant, then the assumption of iid holds. But the Chernoff bound is not tailored for iid. I guess Bernoulli distribution is appropriate here. If this is the case, the reason for using Chernoff bound should be given.

2.      It seems that statistical fluctuations here have greater impact compared with previous protocol. To beat PLOB bound, too many pulses are necessary as illustrated in Fig.1. Then, I wonder the reason behind this, and is there some ways to alleviate the finite-key effects. I think a discussion on this point is helpful.

3.      The analytical formula for estimating n_ph is good. But one may wonder if numerical method leads to better performance. Can the authors compare them?

4.      The parameters (the intensities and corresponding probabilities) in the simulation are optimized at each distance? If so, the algorithm for the optimization should be given.

5.      There are some typos in the manuscript. Please check the manuscript and refine the language carefully.

Reviewer 2 Report

Since there is no security proof for a QKD protocol with discrete phase-randomization in the finite-key region, this manuscript uses the conjugate measurement to propose a security proof for TF QKD. The proposed security proof indicates that TF QKD can perform satisfactorily using a reasonable number of discrete random phases. To the best of my knowledge, the proposed security proof is correct and innovative, and the proposed security proof also has rigorous and careful processes. Therefore, I recommend publishing this manuscript in the journal of Entropy in this version.
